# Ciliary Ultrastructure Assessed by Transmission Electron Microscopy in Adults with Bronchiectasis and Suspected Primary Ciliary Dyskinesia but Inconclusive Genotype

**DOI:** 10.3390/cells12222651

**Published:** 2023-11-18

**Authors:** Ben O. Staar, Jan Hegermann, Bernd Auber, Raphael Ewen, Sandra von Hardenberg, Ruth Olmer, Isabell Pink, Jessica Rademacher, Martin Wetzke, Felix C. Ringshausen

**Affiliations:** 1Department of Respiratory Medicine and Infectious Diseases, Hannover Medical School (MHH), 30625 Hannover, Germany; staar.ben@mh-hannover.de (B.O.S.); ewen.raphael@mh-hannover.de (R.E.); pink.isabell@mh-hannover.de (I.P.); rademacher.jessica@mh-hannover.de (J.R.); 2Biomedical Research in End-Stage and Obstructive Lung Disease Hannover (BREATH), German Center for Lung Research (DZL), 30625 Hannover, Germany; hegermann.jan@mh-hannover.de (J.H.); olmer.ruth@mh-hannover.de (R.O.); wetzke.martin@mh-hannover.de (M.W.); 3European Reference Network for Rare and Complex Lung Diseases (ERN-LUNG), 60596 Frankfurt am Main, Germany; 4Research Core Unit Electron Microscopy, Institute of Functional and Applied Anatomy, Hannover Medical School (MHH), 30625 Hannover, Germany; 5Department of Human Genetics, Hannover Medical School (MHH), 30625 Hannover, Germany; auber.bernd@mh-hannover.de (B.A.); vonhardenberg.sandra@mh-hannover.de (S.v.H.); 6Leibniz Research Laboratories for Biotechnology and Artificial Organs (LEBAO), Department of Cardiothoracic, Transplantation and Vascular Surgery (HTTG), Hannover Medical School (MHH), 30625 Hannover, Germany; 7REBIRTH—Research Center for Translational and Regenerative Medicine, Hannover Medical School (MHH), 30625 Hannover, Germany; 8Department of Paediatric Pneumology, Allergology and Neonatology, Hannover Medical School (MHH), 30625 Hannover, Germany

**Keywords:** bronchiectasis, genotype–phenotype correlation, primary ciliary dyskinesia, transmission electron microscopy, ultrastructure, whole-exome sequencing

## Abstract

Whole-exome sequencing has expedited the diagnostic work-up of primary ciliary dyskinesia (PCD), when used in addition to clinical phenotype and nasal nitric oxide. However, it reveals variants of uncertain significance (VUS) in established PCD genes or (likely) pathogenic variants in genes of uncertain significance in approximately 30% of tested individuals. We aimed to assess genotype–phenotype correlations in adults with bronchiectasis, clinical suspicion of PCD, and inconclusive whole-exome sequencing results using transmission electron microscopy (TEM) and ciliary image averaging by the PCD Detect software. We recruited 16 patients with VUS in *CCDC39*, *CCDC40*, *CCDC103*, *DNAH5*, *DNAH5*/*CCDC40*, *DNAH8*/*HYDIN*, *DNAH11*, and *DNAI1* as well as variants in the PCD candidate genes *DNAH1*, *DNAH7*, *NEK10*, and *NME5*. We found normal ciliary ultrastructure in eight patients with VUS in *CCDC39*, *DNAH1*, *DNAH7*, *DNAH8*/*HYDIN*, *DNAH11*, and *DNAI1*. In six patients with VUS in *CCDC40*, *CCDC103*, *DNAH5*, and *DNAI1*, we identified a corresponding ultrastructural hallmark defect. In one patient with homozygous variant in *NME5*, we detected a central complex defect supporting clinical relevance. Using TEM as a targeted approach, we established important genotype–phenotype correlations and definite PCD in a considerable proportion of patients. Overall, the PCD Detect software proved feasible in support of TEM.

## 1. Introduction

Primary ciliary dyskinesia (PCD) is a rare and heterogeneous genetic multisystem disorder caused by the dysfunction of motile cilia [1]. Its pulmonary manifestation is characterized by persistent retention of hyperconcentrated mucus in the airways, with subsequent chronic inflammation, recurrent upper and lower respiratory tract infections, and progressive lung damage resulting in bronchiectasis [2]. Approximately half of the patients show laterality defects like situs inversus totalis, referred to as Kartagener syndrome. In addition, PCD is associated with extrapulmonary manifestations like male and female subfertility and, rarely, hydrocephalus [1]. The global prevalence of PCD is approximately 1:7500 population [3], although PCD is probably underdiagnosed, in particular among people with bronchiectasis [4].

Unfortunately, there is no stand-alone diagnostic test for PCD and guidelines from the European Respiratory Society (ERS) and the American Thoracic Society (ATS) recommend a combination of tests including measurement of nasal nitric oxide (nNO), high-speed video microscopy (HSVM), immunofluorescence microscopy, transmission electron microscopy (TEM), and genetic analysis [5,6]. However, only TEM and genetics are able to confirm definite PCD by detecting ultrastructural hallmark defects or unambiguous pathogenic or likely pathogenic variants in motile ciliopathy genes with established significance [7,8]. To date, >50 genes have been described with well-known association to characteristic clinical symptoms and ultrastructural defects in TEM, also referred to as genotype–phenotype correlation [1,9]. Genetic analysis by next generation sequencing (NGS) is an expeditious diagnostic that is increasingly used in many research settings [4], but also in the routine work-up of orphan diseases in Germany in the past few years [10]. However, it reveals inconclusive results in approximately 30% of patients with a clinical phenotype compatible with PCD, due to either variants of uncertain significance (VUS; class 3 according to American College of Medical Genetics [ACMG]) in established PCD genes or variants in candidate genes of uncertain significance with assumed but unconfirmed association to motile ciliopathies [1,8].

In the present study, we aimed to assess genotype–phenotype correlations in adult patients with bronchiectasis and clinically suspected PCD, but inconclusive genetics in whole-exome sequencing (WES). In addition to conventional TEM, we assessed the feasibility of the PCD Detect software for evaluating ciliary ultrastructure with regard to the confirmation of outer and inner dynein arm defects by ciliary image averaging.

## 2. Materials and Methods

### 2.1. Patient Population

Between December 2020 and January 2022, we consecutively recruited adult patients with computed tomography-confirmed bronchiectasis from our dedicated clinic at Hannover Medical School, Hannover, Germany. At our center, we routinely use NGS, in terms of WES, for the confirmation of clinically suspected PCD since 2018, while HSVM, immunofluorescence, and TEM have not been readily available in the past due to technical complexity and limited staff resources. Thus, all included patients had clinically suspected PCD with unconfirmed diagnosis and were offered genetic testing by WES within the routine work-up for the etiology of bronchiectasis. We obtained informed and written consent before WES according to national legislation. After WES, all patients included in this study had inconclusive genetic results and did not meet the criteria of definite PCD according to ERS or ATS guidelines [5,6]. They were included regardless of the expected TEM outcome, i.e., even if the gene, in which the respective variants were detected, was presumably associated with normal ultrastructure. Patients had either compound heterozygous or homozygous VUS in established PCD genes, e.g., *DNAH5*, or likely pathogenic compound heterozygous or homozygous variants in PCD candidate genes, e.g., *NME5* [8]. Candidate genes were genes of uncertain significance that at the time of patient recruitment were suspected to play a role in the pathogenesis of motile ciliopathies based on supporting data like genomic location or function, but which were still unconfirmed in human patients [8] or merely had few reported cases [11,12,13]. We included one subject with confirmed PCD due to known pathogenic variants in *DNAH5* and one healthy subject as positive and negative controls, respectively. All patients and control subjects provided informed and written consent for study participation before we conducted nasal brush biopsy for TEM. Finally, we grouped the etiology of bronchiectasis into the following categories: definite PCD; probable PCD; possible PCD; or other diagnoses, e.g., asthma. For the purpose of our study, definite PCD required the detection of an ultrastructural hallmark defect by TEM [7], while we diagnosed probable and possible PCD in patients with a suggestive clinical phenotype and at least one supporting abnormal test (or positive familial segregation analysis) or no clearly abnormal test, respectively [14]. Cystic fibrosis (CF) and alpha-1 antitrypsin deficiency were ruled out in all patients as part of their clinical routine. We extracted clinical data, including medical history, signs and symptoms, and findings from routine diagnostics like nNO from our electronic clinical database (FileMaker Pro, Claris International, Cupertino, CA, USA). We evaluated all patients for laterality defects (situs inversus totalis, situs inversus thoracalis, situs inversus abdominalis, left/right isomerism) and excluded patients with current sinonasal or pulmonary exacerbation, epistaxis, hemorrhagic diathesis, and therapeutic anticoagulation.

### 2.2. Whole-Exome Sequencing 

Genomic DNA was extracted from EDTA blood samples with the NucleoMag Blood kit (Macherey-Nagel, Düren, Germany). We used the IDT Exome library kit (xGen, IDT, Leuven, Belgium) for DNA enrichment and library preparation. WES was performed on the Illumina NextSeq 500 sequencer (Illumina, San Diego, CA, USA). We aligned the exome sequence to the reference genome (GRCh37/hg19) with megSAP, version 0.1-710-g52d2b0c (https://github.com/imgag/megSAP (accessed on 15 October 2023)). Genetic variants were visualized and filtered with GSvar, version 2018_04 (https://github.com/imgag/ngs-bits (accessed on 15 October 2023)), IGV [12], version 2.4.14 and with Alamut^®^ visual, version 2.11.0 (Interactive Biosoftware, Rouen, France). For prediction of splicing effects, we used SSF, MaxEntScan and NNSPLICE (integrated part of Alamut v2.11.0, Interactive Biosoftware, Rouen, France). The genetic variants were classified according to the ACMG standards and guidelines for the interpretation of sequence variants (class 1—benign; class 2—likely benign; class 3—variant of unknown significance; class 4—likely pathogenic; and class 5—pathogenic) [8]. First, all patients were screened for variants in well-established PCD genes as well as the Cystic Fibrosis Transmembrane Conductance Regulator (*CFTR*) gene (Appendix A). Additionally, we used a comprehensive virtual gene panel, which contained PCD candidate genes as published by Paff et al. [15].

### 2.3. Transmission Electron Microscopy and PCD Detect Software

Ciliated epithelial cells were obtained by nasal brush biopsy of the inferior nasal meatus with a cytological brush (Gynobrush, Herenz, Hamburg, Germany) at stable state, i.e., in the absence of acute airway infection or sinonasal and pulmonary exacerbation, respectively. Samples were fixed overnight in 150 mM Hepes solution buffer (pH 7.35) containing 1.5% glutaraldehyde and 1.5% formaldehyde. We processed samples to 60 nm ultra-thin sections for TEM as described in Mariani et al. [16]. Ultra-thin sections were observed with a Morgagni 268 Transmission Electron Microscope (FEI, Eindhoven, Netherlands) and images of transverse orientated cilia (cross-sections) were taken using a side-mounted Veleta CCD camera (Olympus SIS, Muenster, Germany). We analyzed >50 ciliary cross-sections from different cells per sample, only assessing cilia with clear structural features and intact membrane. We analyzed and classified the ciliary ultrastructure according to the international consensus guideline BEAT PCD TEM Criteria, with disease-defining class 1 (hallmark) defects, class 2 defects that indicate the diagnosis of PCD in combination with other supporting evidence, or normal ultrastructure [7]. In addition, we used the software PCD Detect, 2020 (downloaded from: https://doi.org/10.5522/04/12327839, last accessed on 28 September 2023), to analyze outer and inner dynein arms on TEM images [17]. PCD Detect is a cilia-averaging tool that generates an averaged image from multiple, manually chosen and layered cutouts, thus reducing background. Nine cutouts, each including a microtubular doublet with outer and inner dynein arms, were manually taken from each ciliary cross-section. At least 5 ciliary cross-sections from different cells, resulting in minimum 45 cutouts, were analyzed in each sample. We performed a reference free averaging using the same search settings as the developers of the software (angle search range: 360°; angle increment size: 1°; search X and Y: 10 pixels; and translational increment size: 1 pixel) [17]. In the present study, we used the color contour map of best-matching features, with yellow and blue indicating low and high electron density, respectively. In a small subset of subjects we assessed the test–retest reliability of TEM analysis on separate samples from nasal brush biopsies of the same individual taken at different occasions.

### 2.4. Statistical Analysis and Ethical Approval

We assessed the patient cohort by descriptive statistics, using medians, interquartile ranges (IQR), and absolute ranges (SPSS, version 27.0, IBM Corp., Armonk, NY, USA). This study was conducted in accordance to the Declaration of Helsinki. The Institutional Review Board of Hannover Medical School approved our research (No. 9831_BO_S_2021). All patients and control subjects provided informed and written consent prior to study inclusion.

## 3. Results

### 3.1. Patient Characteristics

We included 16 patients with inconclusive results in WES due to VUS in established PCD genes (n = 11) or variants in the candidate genes *DNAH1* (n = 2), *DNAH7*, *NEK10* and *NME5* (n = 1, each; Table 1). Among those, seven patients (44%) were female and nine (56%) were male, with a median age of 41 years (IQR 23–53, range 18–63 years). The majority of patients descended from Germany (n = 7), followed by Turkey (n = 5), Russia, Tunisia, Syria, and Kenya (n = 1, each). At inclusion in our study, all patients had bronchiectasis without a definite diagnosis of PCD. Clinically, we graded patients as probable PCD (n = 10), possible PCD (n = 3), idiopathic bronchiectasis (n = 2), and asthma (n = 1). Nasal NO measurements were available from medical records for all but one subject with *Mycobacterium abscessus* infection. Eight patients showed levels below the diagnostic threshold of 77 nL/min. Overall, median nNO production rate was 46 nL/min (IQR 17–238, range 4–290 nL/min). Median FEV_1_ was 54% of predicted (IQR 36–70, range 30–109; Table 1), indicating moderate to severe airflow limitation for the majority of patients.

### 3.2. Transmission Electron Microscopy Analysis

Overall, we detected ultrastructural hallmark (class 1) defects in six patients with probable PCD and inconclusive WES results due to VUS in the established PCD genes *CCDC40* (n = 3; Figure 1), *CCDC103*, *DNAH5*, and *DNAI1* (n = 1, each; Figure 2). In addition, we observed a characteristic central complex (CC; class 2) defect in one subject with a likely pathogenic homozygous variant in the candidate gene *NME5* (n = 1; Figure 3). In detail, TEM revealed disease-defining microtubular disorganization with inner dynein arm (MTD + IDA) defects in three patients with VUS in *CCDC40* (Figure 1). Notably, the homozygous VUS in *CCDC40* in Patient 13 were only discovered following the detection of the MTD + IDA defect with subsequent reevaluation of WES data, as this ultrastructural defect did not match the initially reported variants in *DNAH5*. Moreover, we found disease-defining outer dynein arm (ODA) defects in three patients with VUS in *CCDC103*, *DNAH5*, and *DNAI1* (n = 1, each; Figure 2). Overall, the findings from averaged images of the PCD Detect software showed perfect agreement with those of conventional TEM imaging (Figure 1 and Figure 2). In the *NME5* patient, TEM images showed ciliary cross-sections with normal 9 + 2 structure (in 42 of 119 evaluated cilia, 35%), but also deviant arrangements with the absence of the central pair (9 + 0 structure, 49/119, 41%; and 8 + 0 structure, 13/119, 11%), an additional central pair (9 + 4 structure, 3/119, 3%), and translocated microtubules (8 + 1 structure, 12/119, 10%; Figure 3). Nasal NO measurements showed nNO levels well below the diagnostic threshold of 77 nL/min in all patients with routine data available (Table 1). All patient had a history of early onset of chronic wet cough and chronic sinusitis with nasal polyps. Four patients, in whom we detected characteristic ultrastructural defects, recalled neonatal Respiratory Distress (NRD). One patient with VUS in *CCDC40* and *DNAI1* had Kartagener syndrome, respectively, and one patient with VUS in *CCDC40* had a history of congenital heart but no laterality defect. One female with VUS in *CCDC40* reported a history of infertility (Table 1). There was no history of hydrocephalus reported in these seven patients.

TEM showed normal ciliary ultrastructure in five patients with VUS in the established PCD genes *DNAH11* (n = 2), *CCDC39*, *DNAH8*/*HYDIN*, and *DNAI1* (n = 1, each). In addition, we observed normal ciliary ultrastructure in patients with variants in the PCD candidate genes *DNAH1* (n = 2) and *DNAH7* (n = 1; Figure 4). Again, the findings from averaged images of the PCD Detect software were well in line with those from conventional TEM imaging (Figure 4). Among patients with normal ultrastructure, two patients had probable PCD, three had possible PCD, and three had other diagnoses, including idiopathic bronchiectasis (n = 2) and asthma (n = 1), based on clinical signs and symptoms as well as diagnostic testing results (Table 1). Both patients with probable PCD had chronic upper airway disease, chronic wet cough, and low nNO levels, while one patient had NRD and chronic *Pseudomonas aeruginosa* infection, respectively. Among the three patients with possible PCD, all had chronic upper airway disease and chronic wet cough, but normal nNO levels. One of those patients had a positive family history and a history of parental consanguinity, each, while another patient recalled NRD and two had chronic *P. aeruginosa* infection. All of the three patients with other diagnoses had chronic wet cough and normal nNO levels, while two of them had chronic *P. aeruginosa* infection (Table 1).

We were unable to obtain a robust sample for analysis by TEM and PCD Detect in one patient with a likely pathogenic homozygous variant in the PCD candidate gene *NEK10* (Patient 16), even after repeat nasal brush biopsies on different occasions. Overall, we performed 24 TEM evaluations on separate samples from 16 patients, including repeats due to insufficient initial samples, evaluation of test–retest reliability (n = 3, each), and one positive and one negative control, respectively. Due to initially insufficient sample quality, we had to repeat nasal brush biopsies in three patients. In two of those, repeat sampling yielded robust findings (Patients 1 and 4), while we were unable to obtain sufficient samples in the patient with variants in *NEK10*. Furthermore, we assessed the test–retest reliability of TEM evaluation in a subset of three patients. Of those, two patients with normal TEM had consistently normal results at repeat sampling and TEM evaluation. However, in one patient with homozygous VUS in *DNAI1* (Patient 8), in who we revealed an ODA defect on initial evaluation, the quality of the retest sample was insufficient and precluded repeat evaluation. In order to assure the validity of our TEM evaluation, we included a healthy person as negative control and a person with PCD and known biallelic pathogenic variants in *DNAH5* as positive control, respectively. In both control subjects, TEM showed the expected findings (Figure 1 and Figure 2). 

## 4. Discussion

We analyzed 16 patients with inconclusive results from WES using conventional TEM and averaging of images by the PCD Detect software and, overall, revealed ultrastructural defects in 7 patients (44%). Considering the nine patients with probable PCD and available TEM, this rate was even higher (78%). In six patients the observed ultrastructural hallmark defects confirmed definite PCD. Moreover, we detected a CC (class 2) defect in another patient and, thus, provide supporting evidence for the relevance of genetic variants in *NME5* as the cause of PCD. In summary, our findings highlight the use of TEM to validate the diagnosis of PCD in patients with inconclusive WES results and, in addition, demonstrate the feasibility of the PCD Detect software in support of TEM. 

PCD is a heterogeneous genetic disorder, with reported prevalence rates relating to geographical variability, ethnicity as well as the expertise and availability of dedicated centers providing appropriate diagnostics [18,19]. However, clinical presentation, disease manifestations, comorbidities, and the clinical course of disease may vary according to additional factors, including age (at diagnosis), the gene in which particular genetic variants occur (genotype), and others [20,21]. All 16 patients included in our study had bronchiectasis, clinical features suggestive of PCD, and inconclusive WES results. Based on characteristic signs and symptoms as well as basic diagnostic testing like nNO, we grouped the etiology of bronchiectasis in probable PCD, possible PCD, and other diagnoses. Although the detection of low nNO levels is a sensitive and specific diagnostic test for PCD, other diseases like CF may also show low nNO values [1,22]. However, seven out of nine patients (78%) with probable PCD and low nNO values had ultrastructural (class 1 and 2) defects in the present study. Notably, the two patients with probable PCD and normal ciliary ultrastructure in TEM had VUS in the PCD genes *DNAH11* and *DNAH8*/*HYDIN*, which are known to have normal ciliary ultrastructure, although even subtle ultrastructural defects may be visualized by image averaging or electron tomography [17,23]. *DNAH8* had first been linked to male infertility, but was also described in a patient with PCD [24,25]. Therefore, we did not expect to find ultrastructural defects in these patients. In contrast, in six out of eight patients with normal ciliary ultrastructure in TEM the diagnosis of PCD was a priori considered to be less likely. Of those, three patients had possible PCD and three patients had other diagnoses. All had normal nNO levels. This finding supports the measuring of nNO as a screening test along with characteristic history and clinical symptoms, as recommended by current diagnostic guidelines [5,6]. However, it should be noted that normal nNO levels have been described in people with PCD associated with genetic variants in several specific genes [1], e.g., *NEK10*, as observed in our study (see below).

Beyond the clinical likelihood of PCD, TEM findings depended on the respective genetic variants in affected genes and their genotype–phenotype correlations. All seven patients with ciliary class 1 and 2 ultrastructural defects in TEM had genetic variants in established PCD genes or PCD candidate genes with known genotype–phenotype correlations, including specific ultrastructural defects. In agreement with previous reports, we reproduced ODA defects in three patients with VUS in the PCD genes *CCDC103*, *DNAH5*, and *DNAI1*, respectively [26,27,28]. Moreover, in three patients with VUS in the PCD gene *CCDC40* and another patient with a homozygous variant in the PCD candidate gene *NME5*, we were able to demonstrate MTD + IDA defects [29] and a CC defect [12], respectively. In addition, we observed characteristic genotype–phenotype correlations concerning clinical features of PCD. While respiratory symptoms virtually occur in all people with PCD, laterality defects, infertility, and congenital heart defects are associated to specific PCD genes [1]. In this regard, we observed situs inversus totalis (Kartagener syndrome) in two patients with VUS in *CCDC40* and *DNAI1* and female infertility in another patient with VUS in *CCDC40*, as previously reported by others [26,29,30].

We revealed a CC defect of ciliary ultrastructure in a patient with a likely pathogenic homozygous variant in the PCD candidate gene *NME5*. To date, there is merely one patient reported from South Korea with homozygous nonsense variants in *NME5*, in whom the central pair was reported to be absent in TEM (9 + 0 structure) [12]. In contrast, we also found ciliary cross-sections with normal ultrastructure (9 + 2 structure), but also cross-sections with an additional central pair (9 + 4 structure) or a translocated microtubule (8 + 1 structure). In line with the patient reported by Cho and coworkers, our patient had chronic upper and lower respiratory symptoms and situs solitus [12]. *NME5* encodes for the RSPH23 protein from radial spokes [1,12]. Other PCD genes encoding for radial spoke components like *RSPH9* are also associated to CC defects [31]. To the best of our knowledge, there is no report of laterality defects in patients with (likely) pathogenic variants in PCD genes encoding for radial spoke components, which implies that these do not influence the nodal cilia [32]. Noteworthy, CC defects may also occur as secondary changes following infection. Accordingly, BEAT PCD TEM Criteria define them as class 2 defects [7], which only indicate PCD diagnosis and always require additional diagnostics to confirm disease. Taken together, the characteristic clinical features, the reduced nNO levels, abnormal HSVM, positive familial segregation analysis, and the CC defect observed in our patient, support the hypothesis that genetic variants in *NME5* may cause PCD in humans.

Our findings have several important implications for the diagnostic work-up of PCD. On the one hand, TEM confirmed definite PCD in a considerable proportion of patients with inconclusive WES results in PCD genes with established genotype–phenotype associations. Remarkably, all subjects with definite PCD and available nNO had levels below the diagnostic threshold. Thus, PCD may be reasonably excluded in patients with bronchiectasis and repeatedly normal nNO levels and VUS in PCD genes with known associated ciliary ultrastructural defects as well as reduced nNO, if TEM unequivocally demonstrates normal findings. On the other hand, TEM cannot rule out or confirm PCD in patients with normal ciliary ultrastructure and VUS in PCD genes that are associated with such, e.g., *DNAH11* or *HYDIN*. Following our diagnostic approach with early application of NGS in patients with suspected PCD, this may suggest to perform TEM as a targeted approach in patients with VUS in PCD genes, which are expected to show ultrastructural defects according to already known genotype–phenotype correlations. However, alternative diagnostics are needed in those patients with VUS in PCD genes expected to show normal ciliary ultrastructure, e.g., HSVM in addition to clinical features and nNO for *DNAH11* and immunofluorescence microscopy for *HYDIN* [1]. However, we observed a mutual interaction between genetic testing and TEM in one patient with Kartagener syndrome and VUS in *DNAH5*, in whom the observed ultrastructural defect (MTD + IDA) did not match the expected outcome and triggered genetic reanalysis and, subsequently, detection of homozygous VUS in *CCDC40* that would otherwise have been missed. 

Without doubt, we had to deal with the quality of nasal brush biopsies for use in TEM. This highlights the complexity of the procedure and may limit its use for broad application in routine care. Ultimately, this precluded TEM evaluation in the patient with a likely pathogenic homozygous variant in the PCD candidate gene *NEK10*. Notably, pathogenic variants in *NEK10* have very recently been reported to be associated with situs solitus as well as normal nNO levels, while affecting ciliary growth, mucociliary transport, and thereby causing motile ciliopathy [33]. In our patient, we performed additional familial segregation analysis of the homozygous variant in *NEK10*, including parents as well as affected and unaffected siblings, thus suggesting disease causation. Likewise, in the meantime, we produced additional evidence for the relevance of the detected likely pathogenic homozygous variant in *NME5* [34], which is beyond the scope of the present work. Lastly, our findings confirm that the PCD Detect software is an easy-to-apply and useful tool, in particular when analyzing faintly contrasted ciliary components like the IDA [17]. 

Our study has limitations. We assumed that two heterozygous variants were compound heterozygous (in trans), while definite confirmation from the analysis of parents was available in merely two patients. In addition, the investigator who performed the visual evaluation of ciliary ultrastructure by TEM and PCD Detect (B.O.S.) was not blinded to the findings from WES. Moreover, the assessment of fertility in clinical routine did not follow a standardized approach. Therefore, we may have missed cases of subfertility. Finally, it should be noted that ultrastructural hallmark defects alone are insufficient to reclassify VUS as pathogenic variants, according to ACMG guidelines that require “well-established in vitro or in vivo functional studies supportive of a damaging effect on the gene or gene product” [8]. In this respect, ultrastructural defects are considered to be unspecific for disease-causing variants in a unique PCD gene, as they may theoretically also be associated with other PCD genes, e.g., MTD + IDA defects are associated with *CCDC40*, but also occur in people with variants in *CCDC39* [1].

## 5. Conclusions

In the present study, we established the definite diagnosis of PCD in a considerable proportion of patients with inconclusive results from WES by TEM, thus providing important information on genotype–phenotype correlations regarding VUS in *CCDC40*, *CCDC103*, *DNAH5*, and *DNAI1* as well as a likely pathogenic variant in the PCD candidate gene *NME5*. This may suggest a targeted approach within the diagnostic work-up for PCD, requiring confirmation among a larger patient population from a multicenter study. However, the finding that we detected an ultrastructural hallmark defect in one patient, which reversely informed the interpretation of genetic analysis and resulted in the definite diagnosis of PCD associated with VUS in *CCDC40*, supports its broader application. Further, the PCD Detect software proved to be a feasible tool in addition to the visual analysis of conventional TEM.

## Figures and Tables

**Figure 1 cells-12-02651-f001:**
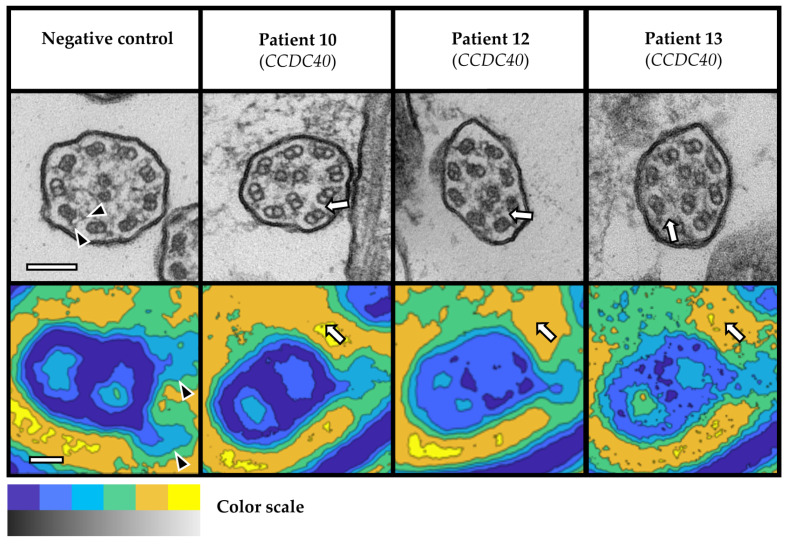
Microtubular disorganization and inner dynein arm (MTD + IDA) defects in three patients with VUS in *CCDC40*. Top: TEM images (greyscale); below: averaged cutout images from the PCD Detect software (colored), representing one microtubule doublet corresponding to the pair oriented downward in TEM. Negative control shows normal ciliary ultrastructure with 9 + 2 arrangement and both dynein arms present (arrowheads). In contrast, arrangement of nine microtubule doublets and a central pair is disorganized and IDAs are absent in the depicted patient samples (arrows). Color scale: Black (TEM) and blue (PCD Detect) as well as white (TEM) and yellow (PCD Detect) indicate high and low electron-density, respectively. Scale bars: 100 nm in TEM images and 10 nm in PCD Detect images. Abbreviations: IDA = inner dynein arm; MTD = microtubular disorganization; TEM = transmission electron microscopy; VUS = variant of unknown significance.

**Figure 2 cells-12-02651-f002:**
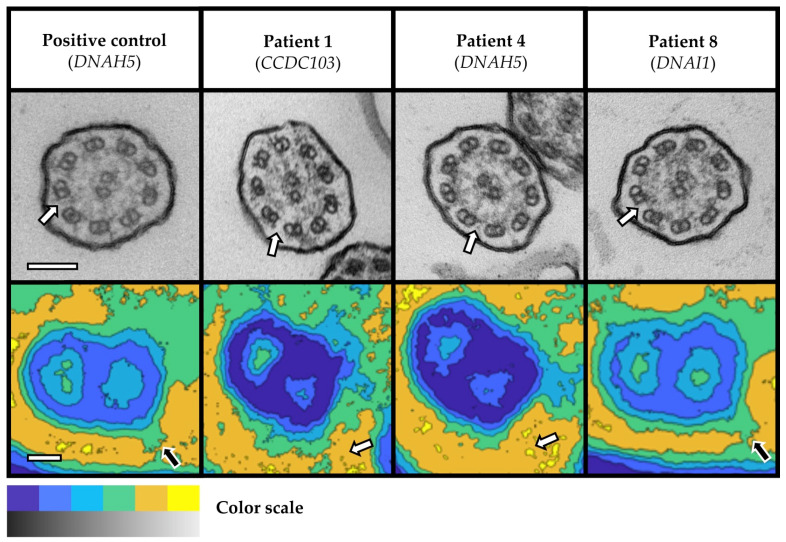
Outer dynein arm (ODA) defects in three patients with VUS in *CCDC103*, *DNAH5*, and *DNAI1*. Ciliary cross-sections in TEM images (greyscale images in upper row) and cutout images of microtubular doublets and ODA (if present) averaged with PCD Detect (colored images in lower row). ODA (white filled arrows) are absent in TEM images of three patients with VUS in *CCDC103*, *DNAH5*, and *DNAI1*, as well as the positive control (*DNAH5*). PCD Detect images of the positive control and Patient 8 show a partial loss of ODA with a remaining shortened ODA projection (black filled arrows), whereas Patients 1 and 4 show a complete loss of ODA (white filled arrows). Color scale: Black (TEM) and blue (PCD Detect) as well as white (TEM) and yellow (PCD Detect) indicate high and low electron-density, respectively. Scale bars: 100 nm in TEM images and 10 nm in PCD Detect images. Abbreviations: ODA = outer dynein arm; TEM = transmission electron microscopy; VUS = variant of unknown significance.

**Figure 3 cells-12-02651-f003:**
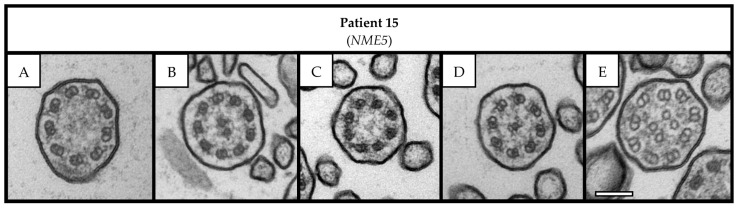
Central Complex (CC) defect in a patient with a likely pathogenic homozygous variant in the PCD candidate gene *NME5.* TEM images show examples of ciliary sections with 9 + 0 structure (**A**), normal 9 + 2 structure (**B**), 8 + 0 structure (**C**), 8 + 1 structure (**D**), and 9 + 4 structure (**E**). Scale bar: 100 nm. Abbreviation: TEM = transmission electron microscopy.

**Figure 4 cells-12-02651-f004:**
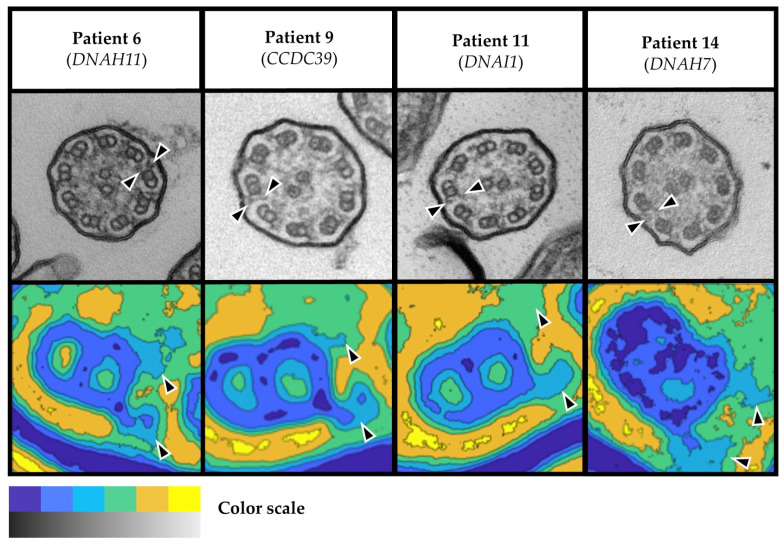
Normal ciliary ultrastructure in four patients with VUS in *CCDC39*, *DNAH11*, *DNAI1*, and *DNAH7*. TEM images of four patients with VUS show normal 9 + 2 structure and both dynein arms present (arrowheads) in conventional TEM imaging (greyscale images in upper row) and PCD Detect imaging (colored images in lower row). Color scale: Black (TEM) and blue (PCD Detect) as well as white (TEM) and yellow (PCD Detect) indicate high and low electron-density, respectively. Scale bars: 100 nm in TEM images and 10 nm in PCD Detect images. Abbreviations: TEM = transmission electron microscopy; VUS = variant of unknown significance.

**Table 1 cells-12-02651-t001:** Individual patient disposition.

Subject	Age	Sex	Origin	Initial Etiology	Etiology after TEM	Nasal NO (nL/min)	ppFEV_1_	Patient History (Signs, Symptoms, and Findings) ^a^	Gene ^b^	cDNA Change	Protein Change	ACMG Class	TEM Results
1	29	F	Turkey	Probable PCD	Definite PCD	11	69	NRD, early onset of chronic wet cough and CRSwNP, repeat sinus surgery, chronic otitis media, and history of grommet insertions	*CCDC103*	c.548T>C c.548T>C	p.(Leu183Pro) p.(Leu183Pro)	3 3	ODA
2	61	F	Russia	Idiopathic bronchiectasis	Idiopathic bronchiectasis	251	37	School-age onset of chronic wet cough, and *P. aeruginosa* infection	*DNAH1*	c.5639C>T c.6925C>A	p.(Thr1880Met) p.(Pro2309Thr)	3 3	Normal
3	63	M	Germany	Possible PCD	Possible PCD	170	33	Pre-school onset of chronic wet cough and CRSwNP, positive family history, history of infertility, and *P. aeruginosa* infection	*DNAH1*	c.2995C>T c.7633A>G	p.(Arg999Cys) p.(Ile2545Val)	3 3	Normal
4	38	F	Germany	Probable PCD	Definite PCD	27	109	NRD, early onset of chronic wet cough, CRSwNP, repeat sinus surgery, chronic otitis media, history of grommet insertions, *P. aeruginosa* infection, and abnormal HSVM (immotile cilia)	*DNAH5*	c.10815delT c.11212-4A>G (intronic)	p.(Pro3606Hisfs*23) -	5 3	ODA
5	58	F	Kenya	Probable PCD	Probable PCD	46	77	Young-adult onset of chronic wet cough, CRSwNP, repeat sinus surgery, and *P. aeruginosa* infection	*DNAH8* *HYDIN*	c.3215G>A c.1169T>C c.12401T>C c.2943T>C c.1043+5G>C	p.(Arg1072Gln) p.(Leu3723Leu) p.(Leu4134Pro) p.(Asp981=) p.?	3 3 3 3 3	Normal
6	18	M	Germany	Asthma	Asthma	240	59	Adolescent onset of chronic wet cough, and *P. aeruginosa* infection	*DNAH11*	c.5924+12G>A c.6226G>A	p.? p.(Val2076Met)	3 3	Normal
7	47	M	Germany	Probable PCD	Probable PCD	37	69	NRD, early onset of chronic wet cough and CRSwNP, repeat sinus surgery, history of infertility, and abnormal HSVM (static stroke)	*DNAH11*	c.7456A>G c.9815>G	p.(Thr2486AIa) p.(Asn3272Ser)	3 3	Normal
8	46	F	Turkey	Probable KS	Definite KS	-	30	Situs inversus, early onset of chronic wet cough and CRSwNP, history of lower lobe resection, middle lobe atelectasis, and *M. abscessus* infection	*DNAI1*	c.1168G>A c.1168G>A	p.(Asp390Asn) p.(Asp390Asn)	3 3	ODA
9	23	M	Tunisia	Possible PCD	Possible PCD	290	36	NRD, early onset chronic wet cough and CRSwNP	*CCDC39*	c.1363-3delC c.1781C>T	p.? p.(Thr594Ile)	3 3	Normal
10	53	M	Germany	Probable PCD	Definite PCD	8	56	NRD, early onset of chronic wet cough and CRSwNP, repeat sinus surgery, chronic otitis media, hearing loss and aid, history of middle lobe resection, and atrial septal defect	*CCDC40*	c.3129delC c.3354C>A	p.(Phe1004Serfs*35) p.(Tyr1118*)	5 3	MTD + IDA
11	19	M	Germany	Idiopathic bronchiectasis	Idiopathic bronchiectasis	236	82	Adolescent onset of chronic wet cough	*DNAI1*	c.1055A>G c.1207C>T	p.(Tyr352Cys) p.(His403Tyr)	3 3	Normal
12	44	F	Germany	Probable PCD	Definite PCD	21	40	Early onset chronic wet cough and CRSwNP, repeat sinus surgery, history of infertility, and *P. aeruginosa* infection	*CCDC40*	c.1345C>T c.2597A>G	p.(Arg449*) p.(Asn866Ser)	5 3	MTD + IDA
13	24	M	Turkey	Probable KS	Definite KS	17	51	Situs inversus, NRD, early onset of chronic wet cough and CRSwNP, repeat sinus surgery, chronic otitis media, *P. aeruginosa* infection, and abnormal HSVM (reduced amplitude, rigid stroke)	*DNAH5* *CCDC40*	c.358G>A c.3656G>A c.615G>C c.615G>C	p.(Asp120Asn) p.(Arg1219His) p.= p.=	3 3 3	MTD + IDA
14	51	M	Turkey	Possible PCD	Possible PCD	112	32	Early onset of chronic wet cough and CRSwNP, parental consanguinity, and *P. aeruginosa* infection	*DNAH7*	c.12056_12060delTATGT c.12056_12060delTATGT	p.(Leu4019Serfs*3) p.(Leu4019Serfs*3)	3 3	Normal
15	23	F	Turkey	Probable PCD	Probable PCD	4	70	Early onset of chronic wet cough and CRSwNP, repeat sinus surgery, history of middle and lower lobe resection, parental consanguinity, positive familial segregation analysis, *P. aeruginosa* infection, and abnormal HSVM (uncoordinated and circular beating)	*NME5*	c.415delA c.415delA	p.(Ile139Tyrfs*8) p.(Ile139Tyrfs*8)	4 4	CC
16	27	M	Syria	Probable PCD	Probable PCD	238	42	NRD, early onset of chronic wet cough and CRSwNP, chronic otitis media, lower lobe atelectasis, affected sibling / positive familial segregation analysis, parental consanguinity, and *P. aeruginosa* infection	*NEK10*	c.1389C>A c.1389C>A	p.(Tyr463*) p.(Tyr463*)	4 4	N/A

^a^ All patients had computed tomography-confirmed bronchiectasis. All patients had situs solitus, unless otherwise mentioned. Only clinical features that were present are mentioned above. ^b^ Gene transcripts to which variants refer: *CCDC39* (NM_181426.2), *CCDC40* (NM_017950.4), *CCDC103* (NM_213607.3), *DNAH1* (NM_015512.5), *DNAH5* (NM_001369.3), *DNAH7* (NM_018897.3), *DNAH8* (NM_001206927.2), *DNAH11* (NM_001277115.2), *DNAI1* (NM_012144.4), *HYDIN* (NM_001270974.2), *NEK10* (ENST00000295720.6); *NME5* (NM_003551.3). Abbreviations: ACMG = American College of Medical Genetics and Genomics; CC = central complex defect; cDNA = complementary DNA; CRSwNP = chronic rhinosinusitis with nasal polyps; IDA = inner dynein arm defect; KS = Kartagener syndrome; MTD = microtubular disorganization; NO = nitric oxide; N/A = not assessable (due to insufficient samples; after repeat sampling); NRD = neonatal respiratory distress; ODA = outer dynein arm defect; PCD = primary ciliary dyskineasia; ppFEV_1_ = percent predicted of forced expiratory volume in one second; TEM = transmission electron microscopy. Explanations of symbols used with protein change: According to the Sequence Variant Nomenclature of the Human Genome Variation Society (version 20.05; https://varnomen.hgvs.org; accessed on 15 October 2023), “p.=” means that the entire protein coding region was analyzed and no variant was found that changes (or is predicted to change) the protein sequence; “*” means that the amino acid is changed to a stop codon; and “?” means that the consequence, at the protein level, of a variant affecting the translation initiation codon cannot be predicted, i.e. is unknown.

## Data Availability

Data are available from the authors upon request.

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
