# Peer review of "Ciliary Ultrastructure Assessed by Transmission Electron Microscopy in Adults with Bronchiectasis and Suspected Primary Ciliary Dyskinesia but Inconclusive Genotype"

_cells, 2023, doi:10.3390/cells12222651_

Round 1

Reviewer 1 Report

Comments and Suggestions for Authors

Background: Primary ciliary dyskinesia (PCD) is caused by dysfunctional motile cilia and is a rare

genetic disorder that presents clinicians with diagnostics challenges resulting in underdiagnosis

especially in patients with bronchiectasis. This paper highlights the challenges with whole exome

sequencing (WES) as it reveals variants of uncertain significance (VUS) in established PCD genes

even when used alongside clinical phenotype and nasal nitric oxide (nNO) as a diagnostic work-up in

PCD. The findings of this paper show important genotype-phenotype correlations and 35 definite PCD

in a considerable proportion of patients with bronchiectasis using Transmission electron microscopy

(TEM) as a targeted approach.

This study is a first to provide definite diagnosis of PCD in a considerable patients with inconclusive

results from WES using TEM. It highlights important information on genotype-phenotype correlations

regarding a likely pathogenic variant in the PCD candidate gene NME5 as well as variant of unknown

significance (VUS) in CCDC40, CCDC103, DNAH5, and DNAI1 as well as.

Summary of key results:

TEM analysis showed normal ciliary ultrastructure in 8 patients with VUS in CCDC39, DNAH1,

DNAH7, 31 DNAH8/HYDIN, DNAH11, and DNAI1. Whereas, ultrastructural defects were observed in

6 patients with VUS in CCDC40, CCDC103, DNAH5, and 32 DNAI1. A homozygous variant in NME5

showed a central complex defect supporting clinical relevance in PCD.

Overall summary:

Aside from clinical aspects, TEM findings relied on the genetic variants in affected genes and their

genotype-phenotype correlations which led to better diagnosis of PCD in cases where WES findings

were inconclusive. Findings from this study can be used in early application of NGS in patients with

suspected PCD along with performing TEM as a targeted approach in patients with VUS in PCD

genes, which are expected to show ultrastructural defects according to already known genotype-

phenotype correlations. Study also shows that the PCD Detect software is an easy-to-apply and

useful tool, in particular when analyzing faintly contrasted ciliary components like the inner dynein arm

(IDA).

Critique:

Strengths:

1) Authors have done a good job at consolidating all the data showing the consistency between

the TEM findings with that of PCD detect software.

2) Experimental design is quite robust with adequate number of samples subjected to test-retest

analysis with both positive and negative controls showing expected TEM findings.

3) Authors rightly point out the presence of diseases like cystic fibrosis in the background

resulting in low nNO levels which need not be attributed to PCD only.

4) Female infertility and situs inversus totalis detected in VUS in line with findings from other

studies.

5) Authors rule out hydrocephalous in 7 patients with VUS in CCDC40 which is in line with the

findings from other papers that does not associate hydrocephaly with CCDC40 mutation.

(Becker-Heck. A, 2011)

6) An important finding is a CC defect of ciliary ultrastructure in a patient with a likely pathogenic

homozygous variant in the PCD candidate gene NME5.

7) Strength of this study lies in the TEM data showing conclusive PCD in cases with uncertain

WES results.

8) For better correlation, authors have provided in detail information on nNO levels and FEV1

which helps to know where on the spectrum are the PCD patients as far as disease severity

was concerned.

Weaknesses:

1) Title does not reflect the significance of methods that are the focus of this paper, need to rephrase it to include TEM.

2) TEM analysis of ciliary ultrastructure was carried out on 16 PCD patients with which needs to be extended to larger patient population to rule out any variability due to sampling bias

3) This paper does not discuss the overall prevalence rate, life-expectancy, co-morbidities as well as other mutation types commonly observed in PCD.

4) There are confounding factors as far as the nNO levels are concerned as not all PCD confirmed cases show abnormal nNO values. Authors might want to highlight other factors resulting in PCD despite normal nNO levels.

5) Authors recommend nNO testing as a reliable tool despite not having a definitive data supporting it and so a significance level needs to be determined in cases with confirmed PCD diagnosis wherever abnormal nNO levels were observed.

6) It would be better to include history of infertility and congenital heart defects in the individual patient disposition table.

7) Authors do not mention how the average was taken while analysing the normal as well as abnormal ciliary ultrastructure. Number of ciliary structures (n) taken into account is not clearly stated.

8) Authors state that laterality defects might not influence nodal cilia but its not mentioned how many patients with likely pathogenic variants in PCD gene were evaluated for it.

9) Authors state that PCD maybe excluded in patients with normal nNO levels, however, the

result section shows PCD diagnosis in 3 patients that had normal nNO levels and 3 other

patients with abnormal nNO levels so this needs to be explained.

10) Lack of blinding on the investigator performing TEM and PCD test to WES findings weaken

the strength of the evidence around the reliability of such tests in PCD diagnosis.

11) One drawback of the TEM technique here is the quality of nasal brush biopsies due to complexity of the procedure resulting in retrieving insufficient amount of sample.

Reviewer 2 Report

Comments and Suggestions for Authors

In this manuscript entitled “Ciliary Ultrastructure in Adults with Bronchiectasis and Suspected Primary Ciliary Dyskinesia but Inconclusive Genotype”, authors disclose that several variants of PCD established and candidate genes cause ciliary ultrastructural defects via TEM images assisted with PCD Detect Software. This manuscript would truly expand our understanding of the genetic spectrum of PCD through these various gene loci, however, this work does not pay attention to and discuss the different clinical traits/cellular changes/ultrastructure variations among the reported variants and these variants in this manuscript. If these limitations are overcome, the quality of this manuscript will reach a new level.

In general, there are still some concerned issues for careful consideration to make this manuscript more readable, clear, and credible.

1. In this manuscript, authors need to carefully examine the style of writing a gene or protein name. e.g., Line 29 in the abstract section, DNAI1 should be in Italic style to represent a gene name.

2. No phenotypic descriptions of NEK10 variant in ciliary ultrastructure appear that could be deleted in this manuscript. The keynote of this manuscript is to elucidate the ciliary ultrastructure underlying the different PCD variants.

3. Please describe how to group “probable PCD” and “possible PCD” according to certain criteria.

4. In this manuscript, the authors mainly used TEM and PCD Detect Software to analyze the ultrastructural ciliary defects. I don't quite understand how this PCD Detect Software works and gives us the averaging image ranging from yellow (low densities) to blue (high densities). Especially, it looks that the normal structure of outer doublet microtubules is assigned as various color densities, e.g., Fig. 2 (light blue and dark blue both for normal MT doublet). For this reason, is it a challenge to reflect the normal structures and defective structures in this format of data presentations.

5. Line 190-191, TEM images show that mutation of DNAI1 causes ODA defects. However, in Line 32, the authors have stated that normal structures are found in the cilia of patients with DNAI1 mutation. Is it correct? If variants in the same DNAI1 gene finally lead to distinct phenotypes, should authors execute additional assays (e.g., WB or IF) to detect and analyze the expression level or localization of DNAI1 variant proteins in a cell, or analyze whether the mutated amino acid sites are important for the functional wild-type protein.

6. As the authors stated in this manuscript in the Introduction section, various methods should be applied for the diagnosis of PCD. Is it possible and applicable to check the cellular cilia via IF or protein expression level via WB from these patient cells, and then analyze these data by scatter graph depicting data such as the ratio of ciliation/ciliary length/intensities of certain ciliary protein/…..

7. Species names need to be written in Italic style. e.g., Line 261.

8. Are the titles and subtitles appropriate in the Results section?

9. As the author states in this manuscript, all the listed mutated genes are reported in other papers. If possible, the authors could explain some new findings that differ from other case reports. Indeed, it is highly interesting to investigate clinical and cytopathic mechanisms in those variants of known PCD genes and variants of PCD candidate genes in future studies. Functional analysis of different PCD variants of proteins will help us to understand the PCD spectrum and devise a targeted therapy strategy.

10. The title should be reconsidered with a more suitable one for this full text.

The genotype-phenotype correlations reported in this manuscript by authors certainly expand our knowledge about the PCD mutagenesis spectrum, however, it does not seem to comparatively indicate that their findings of these new loci in this text are different from reported loci. Meanwhile, the presented data/figures are relatively insufficient as an article type in my opinion.

Overall, this manuscript should be thoroughly revised to make better clarity and scientificity of text presentations according to the suggestions below. If possible, authors could provide additional data/analysis to further enhance the integrity of the manuscript, and also truly improve our understanding of the genetic spectrum of PCD.

Reviewer 3 Report

Comments and Suggestions for Authors

This is a clear, well written report describing genetic and nasal cilia EM findings in a small cohort of adult bronchiectasis patients. I find the topic important, likely PCD is gretaly underdiagnosed in adult bronchiectasis patients as in contrast to pediatric individuals, often no testing is performed.

I have a few suggestions 

1. Introduction:  IF is also of great help in PCD diagnostics and could be mentioned here. Further, this should be mentioned in the discussion, e.g. when TEM limitiations are discussed. 

2.  results patient description: PICADAR scores of the patients could be of additional interest

3. Patient 10 carries 2 nonsense alleles, why is this genetically inconclusive? 

4. NME5 case: the authors describe in the discussion section that they found cross sections with normal ultrastructure as well as with 9+0, 9+4, 8+1...which were the frequencies of those abnormalities? A range of defects can also occur as secondary effect after infection, checiking for defects after culturing or in a second brushing in an infection free interval may be helpful

5. discussion: of the cases with normal ciliary ultrastructure in EM, a large proportion had nasal NO values above the PCD cutoff while many of the cases with low nNO had indeed ultrastructural defects. This suggests that the likelyhood for PCD is high with lo nNO (that^s expected as the cutoff was chosen for this purpose...) and it is low with nNO above the PCD cutoff (also expected and desired) in adult brnochiectasis. this should be made clear in the discussion section, by follwoing the nNOcutoff, most PCD/non-PCD classifications would have been correct. 

Round 2

Reviewer 2 Report

Comments and Suggestions for Authors

The authors have explained my concerns in the response letter and revised the initial manuscript. As mentioned by the authors in the response letter, TEM along with the PCD Detect Software technique should be a fast and concise clinical routine setting for uncovering PCD loci. This protocol will truly accelerate clinical diagnosis and early intervention of PCD.

Agree to publish in Cells journal.